# *TSSKL* is essential for sperm mitochondrial morphogenesis and male fertility in moths

Xia Xu[1]☯, Lu Zhu[2,3]☯, Xiaomiao Xu[3], Jine Chen[1], Xin Du[1], Linbao Zhu[1], Shaofang Yu[1], Lansa Qian[4], Xingchuan Jiang[3], Lijun Zhou[3], Yongcheng Dong[3], Yongqiang Wang[1]*, Yongping Huang[2]*, Yaohui Wang[3,4]*

1 Institute of Sericulture and Tea, Zhejiang Academy of Agricultural Sciences, Hangzhou, China, 2 School of Environmental Science and Engineering, Shanghai Jiao Tong University, Shanghai, China, 3 Key Laboratory of Agricultural Product Quality and Biosafety of the Ministry of Education, College of Plant Protection, Anhui Agricultural University, China, 4 Key Laboratory of Plant Design, CAS Center for Excellence in Molecular Plant Sciences, Chinese Academy of Sciences, China

☯ These authors contributed equally to this work.
* insectgroup@sjtu.edu.cn (YPH); wangyq@zaas.ac.cn (YQW); wyhhyf@163.com (YW)

## Abstract

Sperm deliver male genomic DNA to the ovum, playing a pivotal role in sexual reproduction across the animal kingdom. The molecular regulation of sperm morphogenesis has consequently become a focal point of genetic research, with dual implications for both reproductive medicine and sustainable agriculture. Here, we characterize the functional role of the testis-specific serine/threonine protein kinase-like (*TSSKL*) gene in the model lepidopteran insect *Bombyx mori* and the globally destructive crop pest *Plutella xylostella*. RNA-seq and qPCR analyses revealed *TSSKL*'s testis-specific expression pattern. Using CRISPR/Cas9-mediated mutagenesis, we demonstrate that *TSSKL* knockout induces complete male sterility, while female fertility remains unaffected compared to wild-type. Fluorescence microscopy and ultrastructural analyses revealed that *TSSKL* deletion leads to severe morphological defects in both eupyrene and apyrene sperm, accompanied by impaired mitochondrial dynamics and aberrant autophagy. Comparative transcriptome and functional analyses linked these phenotypes to dysregulated energy metabolism pathways. Crucially, this sterility phenotype is conserved in *P. xylostella*, recapitulating the findings in *B. mori*. Our study demonstrates that *TSSKL* is crucial for male fertility, coordinating both structural and metabolic aspects of sperm development. Beyond advancing fundamental knowledge of insect reproductive biology, this work also identifies *TSSKL* as an ideal target for lepidopteran pest control through precision sterility induction.

## Author summary

As the exclusive vectors of paternal genetic material, sperm are indispensable for sexual reproduction. Consequently, sperm morphogenesis has emerged as

**Data availability statement:** All relevant data are within the paper and its Supporting Information files (including raw data).

**Funding:** This work was supported by the National Natural Science Foundation of China (32202283 to Y.H.W. and 32100377 to X.X.), Technological Grant of Zhejiang for Breeding New Agricultural Varieties (2021C02072 to Y.Q.W.), Key Scientific and the China Agriculture Research System of MOF and MARA (CARS-18 to X.X.), Education Department of Anhui Province Outstanding Youth Project (2023AH030045 to Y.H.W.), Talent Research Project of Anhui Agricultural University (rc342219 to Y.H.W.), Scientific research projects of universities in Anhui Province (2024AH050442 to X.C.J.), and the Research Foundation of Education Department of Anhui Province, China (KJ2021A0194 to Y.C.D.). The funders had no role in study design, data collection and analysis, decision to publish, or preparation of the manuscript.

**Competing interests:** The authors have declared that no competing interests exist.

a critical research frontier - both for elucidating reproductive biology and developing targeted pest management strategies. In this work, we characterized the functional role of the testis-specific gene *TSSKL* in two lepidopteran species: *Bombyx mori* and *Plutella xylostella*. CRISPR/Cas9-mediated *TSSKL* knockout resulted in complete male sterility, while maintaining normal female fertility, demonstrating its sex-specific reproductive function. Comprehensive ultrastructural and fluorescence microscopy analyses revealed that *TSSKL* deficiency leads to mitochondrial derivative deficiency and activation of autophagy, impairing energy production essential for sperm function. Notably, this sterile phenotype was conserved across both species, indicating *TSSKL*'s fundamental role in lepidopteran sperm morphogenesis. Our findings provide novel mechanistic insights into insect reproductive genetics while identifying *TSSKL* as a promising target for species-specific population control through precision sterility induction.

## Introduction

In most animal species, males typically produce a single sperm morphotype [1,2]. Lepidopterans represent a remarkable exception through sperm dimorphism, producing two functionally distinct sperm types: nucleated eupyrene sperm and anucleate apyrene sperm [3]. During spermiogenesis, these sperm types show distinct developmental patterns: eupyrene sperm form organized axial filaments and mitochondria that assemble into the sperm tail near the centrosomal region, whereas apyrene sperm mitochondria maintain smaller mitochondria with a distinctive "V"-shaped configuration [4,5]. Notably, apyrene sperm display greater motility than eupyrene sperm, and plays an important auxiliary role in the fertilization process [6], with both sperm types being essential for successful reproduction in Lepidoptera. The motility of sperm, a critical factor for fertility, depends on adenosine triphosphate (ATP) flagellar beating [7]. As mitochondria generate the primary source of ATP through oxidative phosphorylation, their proper function is vital for male fertility [8,9]. Consequently, disruptions in mitochondrial activity - including oxidative stress and apoptotic pathways activation - can impair sperm motility and ultimately reduce male reproductive success [10,11].

Protein kinase, also known as protein phosphorylase, is ubiquitous in eukaryotic cells [10]. These enzymes catalyze the transfer of phosphoric acid from ATP and covalently bind to hydroxyl groups of certain serine, threonine, or tyrosine residues in specific protein molecules [12,13]. Thus, the conformation and activity of proteins and enzymes are changed [11]. Serine/threonine-protein kinase is a kind of protein kinase family, which plays vital roles in developmental and reproductive processes [14]. Among these kinases, testis-specific serine/threonine-protein kinases (TSSKs) exhibit exclusive testicular expression and regulate critical aspects of spermatogenesis and sperm motility [15,16]. Six conserved *TSSK* genes (*TSSK1–6*), all containing the characteristic S_TKc domain, have been identified [17,18]. These kinases demonstrate specialized functions: *TSSK1* and *TSSK2* are expressed exclusively in

the testes during late sperm maturation in sexually mature individuals, and their deficiency results in male sterility [19,20]. *TSSK3* shows continuous post-meiotic expression in testes, with its deficiency causing sperm morphological defects [21]. *TSSK4* regulates sperm motility, and its knockout leads to sperm flagellar bending [22]. *TSSK5* phosphorylates cAMP response element-binding protein (CREB) to facilitate spermatogenesis [23,24], whereas *TSSK6* deletion disrupts actin polymerization, resulting in abnormal sperm morphology and male sterility [25,26]. Despite extensive characterization of TSSKs in lepidopterans, the understanding of specific factors involved in reproduction remains limited, particularly in regard to the molecular mechanisms underlying processes such as sperm mitochondrial morphogenesis and sperm migration.

In this study, we utilized CRISPR/Cas9 to investigate *TSSKL* gene function in the model lepidopteran *B. mori*. *BmTSSKL* knockout induced complete male sterility while maintaining normal female fertility. Immunofluorescence and transmission electron microscopy analyses demonstrated *BmTSSKL*'s essential role in both eupyrene and apyrene sperm development, particularly in maintaining mitochondrial ultrastructure integrity. RNA-seq revealed a reduction in mitochondrial metabolism in the testes of *BmTSSKL* mutant males. Remarkably, these phenotypic effects were conserved in the agricultural important lepidopteran pest *Plutella xylostella*. Our results provide mechanistic insights into TSSKL's essential role in lepidopteran sperm morphogenesis while highlighting its potential as a molecular target for pest control strategies.

## Results

### *BmTSSKL* exhibits testis-specific expression in *B. mori*

Given the testis's crucial role in sperm morphogenesis and sperm maturation [27], we compared the transcriptomes of the testes, ovaries, whole body of 2 days old virgin adult excluding the testis or ovary (Fig 1A). Comparative analysis revealed 1293 uniquely expressed genes in testes compared to other three organs (Fig 1B). Remarkably, testicle-specific serine/threonine protein kinase-like (*TSSKL*, accession number: PX116274) was significantly enriched in testes (S1 Table). We additionally performed qRT-PCR analysis of 10 different tissues of *B. mori* adults and found that *BmTSSKL* was predominantly expressed in the testis than other tissues analyzed (i.e., head, epidermis, fat body, malpighian tubules, midgut, middle silk gland, posterior silk gland, anterior silk gland, and ovary) (Fig 1C). Furthermore, we quantified *BmTSSKL* expression pattern in the testis from the fifth-instar larvae stage through the adult stage and observed that it gradually increased during development, reaching a higher level at the virgin adult stage (Fig 1D). These results suggested that the *BmTSSKL* played a potential role in the regulation of sperm morphogenesis.

### *BmTSSKL* mutations cause male-specific sterility without affecting mating ability

Based on the *B. mori* genome references [28], we found that *BmTSSKL* is located on chromosome 25 and only has one exon (Fig 1E). To explore the physiological function of *BmTSSKL*, we used the binary transgenic CRISPR/Cas9 system to engineer loss-of-function mutants. To do this, we created a transgenic silkworm line expressing the two sgRNA targeting *BmTSSKL* driven by the silkworm U6 promoter (Fig 1F and 1G). Meanwhile, the insertion of the *U6-sgRNAs* and *Nos-Cas9* transgenes was validated by inverse-PCR performed on genomic DNA isolated from the whole bodies of fluorescent G1 moths (DsRed or GFP positive) (Fig 1G). Analysis of the resulting sequences against the silkworm genome database (http://kaikoblast.dna.affrc.go.jp) thereby localized the transgenes to distinct chromosomal loci (S1 Fig). Subsequently, the *U6-sgRNA* (*TSSKL*) and *Nos-Cas9* lines were crossed to generate the *TSSKL* mutant line (Figs 1G and S2). Genomic DNA was then extracted from randomly selected dual fluorescent (dsRed and GFP) individuals, and the target regions were PCR-amplified and sequenced. Sequencing analysis revealed the occurrence of various large deletions in the *BmTSSKL* transgenic line (Fig 1H). These results demonstrated that we had successfully obtained the *BmTSSKL* deficiency mutants. As predicted, the *BmTSSKL* mutants had viable and normal surface phenotypes compared with WT during all development stages (S3 Fig).

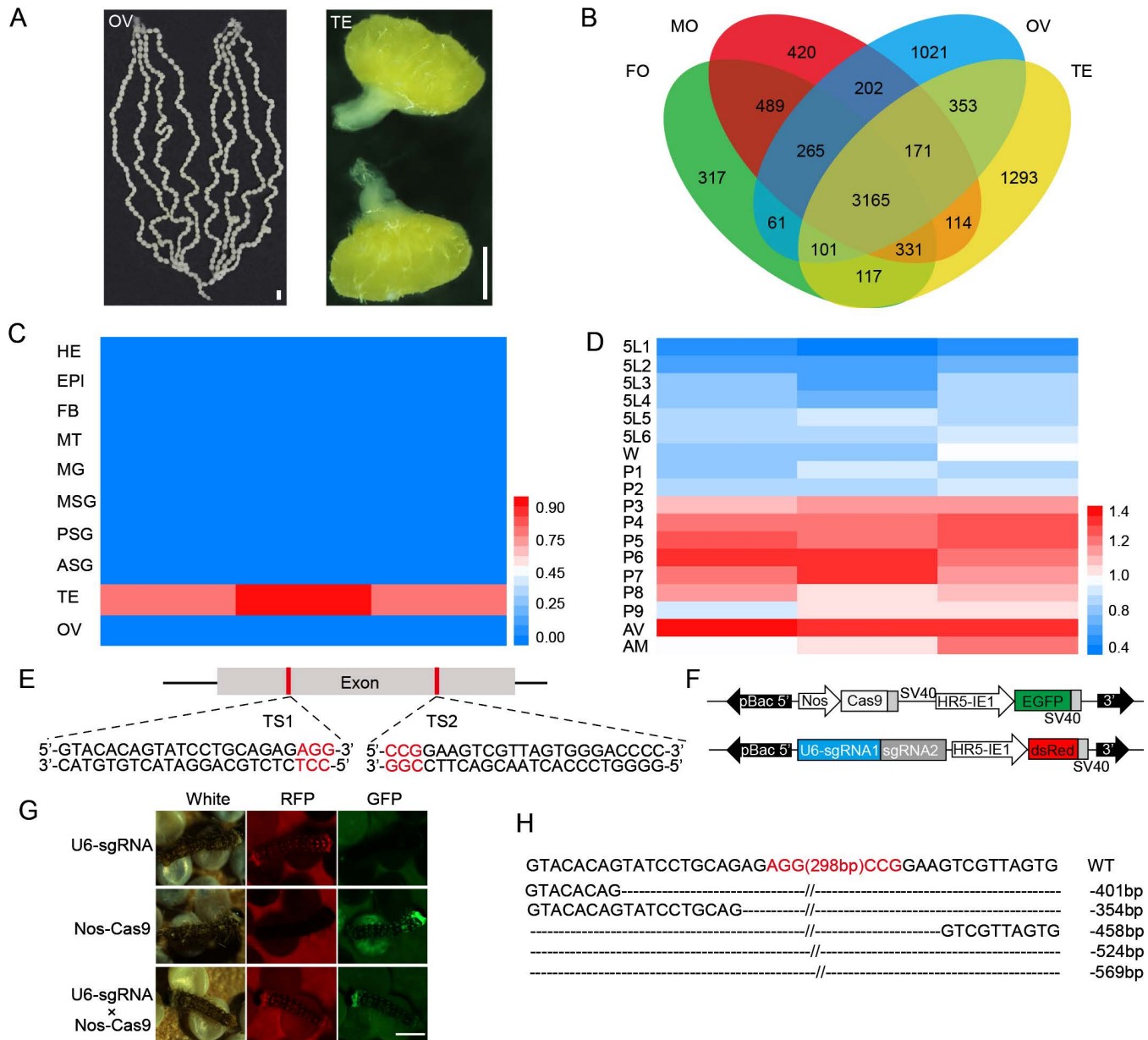

**Fig 1. Spatiotemporal expression pattern and targeted mutagenesis of *BmTSSKL*.** (A) The representative morphology of testis and ovary dissected from virgin adults. Scale bar, 1 mm. (B) The number of specific expression genes in the testes compared with the other tissues. TE, testis; OV, ovary; MO, whole body of 2 days old virgin adult males excluding the testis; FO: whole body of 2 days old virgin adult females excluding the ovary. (C) qRT-PCR analysis of *BmTSSKL* mRNA levels in various tissues from the fourth day of the fifth larval instar. Abbreviations: HE, head; EPI, epidermis; FB, fat body; MT, Malpighian tubules; MG, midgut; ASG, anterior silk gland; MSG, middle silk gland; PSG, posterior silk gland; TE, testis; OV, ovary. The mRNA expression level was normalized to *B. mori ribosomal protein 49* (*Bmrp49*), an internal reference. Three independent replicates were performed. (D) qRT-PCR analysis of *BmTSSKL* mRNA levels in testes at indicated times. 5L1, the first day of the fifth instar larvae; W, wandering; P, pupae; AV, virgin adult; AM, mated adult. Three independent replicates were performed. (E) Genomic structure of *BmTSSKL*. The locations and sequences of the binding sites for sgRNAs (TS1 and TS2) in the exon are indicated. (F) Schematic representations the activator line vector *Nos-Cas9* and effector line vector *U6-sgRNA*. (G) Fluorescent screening of *BmTSSKL* mutant larvae. Scale bar, 1 mm. (H) The sequence of the region between sgRNA target sites in the *BmTSSKL* gene for WT and four randomly selected *BmTSSKL* mutant G1 individuals. Dashed lines represent the deleted bases. The target sequence is in black, and the PAM sequences are in red. The deletion size is indicated to the right of the sequence.

Next, to explore whether the disruption of *BmTSSKL* affects the fertility of *B. mori*, we performed fecundity assays. *BmTSSKL* mutant males mated with either WT or *BmTSSKL* mutant females produced normal egg yields, but these eggs failed to hatch (0%). Whereas WT males mated with either WT or *BmTSSKL* mutant females produced normal egg yields with successful hatching (94–96%) (Fig 2A). We also counted the number of eggs in broods produced by different cross combinations. The controls produced an average of ~370 eggs (WT: 370, *U6-sgRNAs*: 370, *Nos-Cas9*: 369); WT males mated with *BmTSSKL* mutant females produced average 368 eggs; *BmTSSKL* mutant males mated with WT females produced average 348 eggs; *BmTSSKL* mutant males mated with *BmTSSKL* mutant females produced average 339 eggs (Fig 2B). After free hybridization of *Nos-Cas9* and *U6-sgRNAs*, four types of offspring were produced, including dual fluorescent *BmTSSKL* mutant line, *Nos-Cas9* line (GFP), *U6-sgRNAs* line (DsRed), and non-fluorescent line (S2 Fig). Subsequently, random hybridization among these four lines only yields offspring of these same four types. Female mutants carrying the male sterility factor could mate non-mutant adult individuals and effectively reproduced the next generation. In this next generation, male mutants could mate but were sterile, while female mutants could both mate and remain fertile. In every generation of silkworms, *BmTSSKL* mutant male sterility was observed and inherited (Fig 2C). Meanwhile, we performed immunofluorescence analysis on the testes of virgin adults and found that there was almost no TSSKL protein in the mutants (Fig 2D). We furthermore evaluated the mating behaviors of *BmTSSKL* mutant moths. Behavior was considered normal if females and male moths could attract each other and copulate for at least 30 min [29]. The response index was determined as the percentage of successful mating relative to the total trials in a group (Fig 2E and 2F). There were no significant differences in female attractiveness between *BmTSSKL* mutant and control individuals (WT female: 52.07%, mutant female: 47.93%, $P = 0.262$; *U6-sgRNAs* female: 50.83%, mutant female: 49.17%, $P = 0.448$; *Nos-Cas9* female: 51.00%, mutant female: 49.00%, $P = 0.429$) (Fig 2G). And there was no significant difference in male competitiveness between *BmTSSKL* mutant and control individuals (WT male: 52.05%, mutant male: 47.95%, $P = 0.104$; *U6-sgRNAs* male: 49.62%, mutant male: 50.38%, $P = 0.532$; *Nos-Cas9* male: 50.45%, mutant male: 49.55%, $P = 0.771$) (Fig 2H). In addition, we amplified and sequenced the *BmTSSKL* gene from the testis of injected males with abnormal fertility and the ovary of injected females with normal fertility. Heterozygous chromatograms and various deletions or insertions were observed in the *BmTSSKL* mutant individuals of both sexes (S4 Fig). Taken together, these results demonstrate that *BmTSSKL* is essential for male fertility, but does not affect mating behavior.

### *BmTSSKL* is essential for sperm morphogenesis and sperm migration

We performed further analyses to assess the cause of *BmTSSKL* mutant male sterility. We first investigated whether the *BmTSSKL* mutant males exhibited any gross defects in external genitalia or internal genitalia system. However, we confirmed that no significant defects were detected. In lepidopteran insects, the bursa copulatrix consists of proteins from the seminal fluid that are transferred from male moths during copulation [30]. We further explored whether the *BmTSSKL* mutant influences bursa copulatrix formation. For this purpose, we evaluated the bursa copulatrix 1 h post copulation (hpc) in WT females mated with WT or *BmTSSKL* mutant males. To our surprise, the results showed that the bursa copulatrix was full in WT females mated with WT or *BmTSSKL* males (Fig 3A). We then investigated whether *BmTSSKL* mutant affects sperm transfer into spermatheca. We found that at 3 hpc the spermathecae of females mated with WT males contained sperm, whereas the spermathecae of WT females mated with *BmTSSKL* mutant males were empty, like those of virgin females (Fig 3B). To elucidate *BmTSSKL* is involved in sperm migration in the bursa copulatrix, we compared mRNA expression levels of sperm motility-related genes (*BmMlc* [31], *BmMlc2* [32], *BmTry* [33], and *BmFln* [34]) at different developmental stages between wild-type and *BmTSSKL* mutant individuals using qRT-PCR analysis. Notably, the mRNA levels of *BmMlc*, *BmMlc2*, *BmTry* and *BmFln* were significantly lower in *BmTSSKL* mutant than those of WT (Fig 3C).

In addition, to gain insights into whether *BmTSSKL* is involved in sperm morphogenesis, we performed fluorescence staining to examine the morphology of eupyrene and apyrene sperm bundles in WT and *BmTSSKL* mutant newly

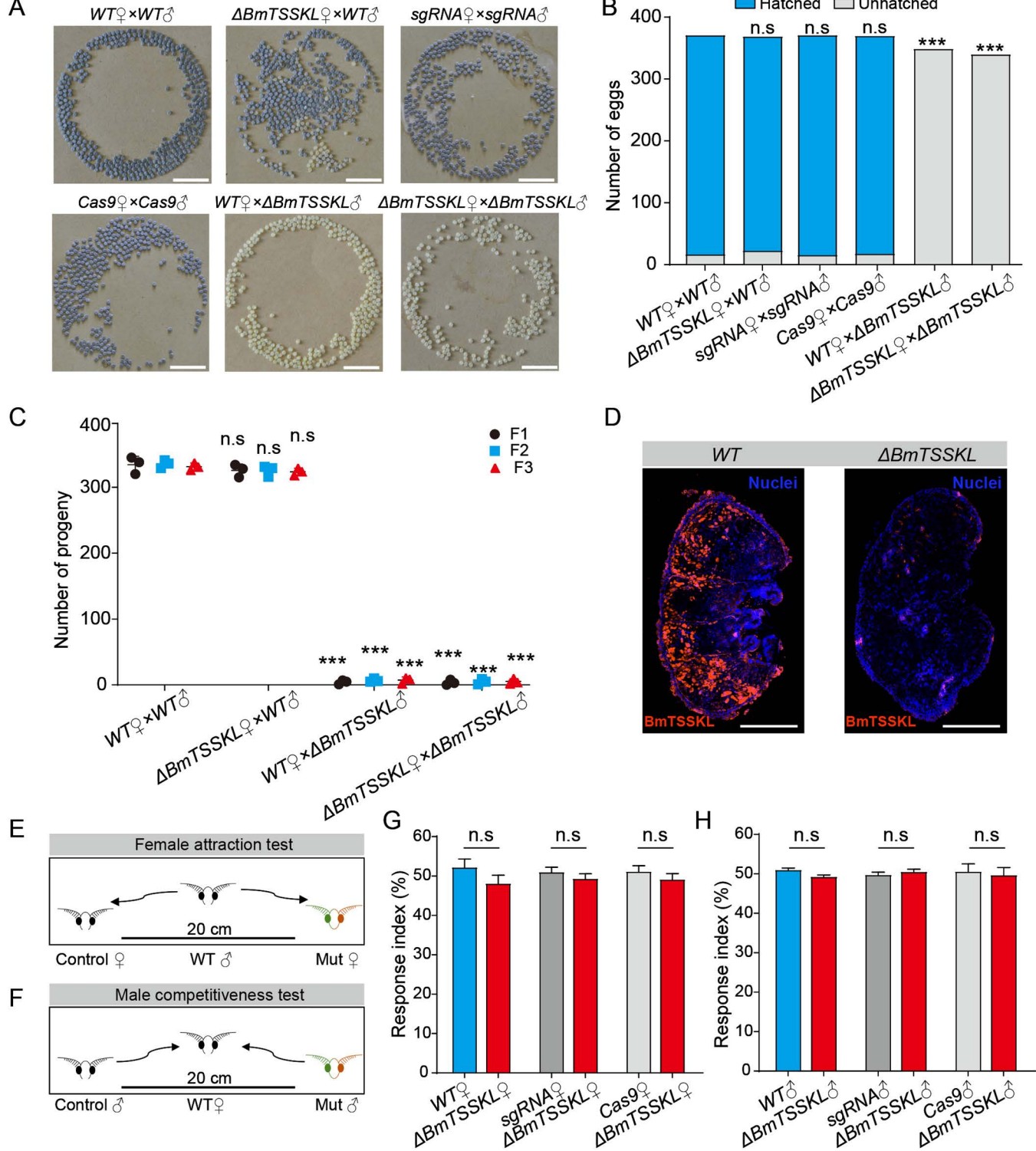

**Fig 2. Mutation of *BmTSSKL* results in male sterility.** (A) Representative photographs of eggs produced by different crosses. Eggs were photographed 8 days after spawning, 2 days before hatching. Developing eggs are dark and undeveloped embryos are light. Scale bar, 1 cm. (B) Plot of the number of eggs hatched and unhatched resulting from different crosses. The experiments were performed three times. Each group has statistically analyzed at least 30 pairs of moths (n > 30). The data shown are means ± S.E.M. Asterisks indicate significant differences with One-way ANOVA test:

***$P<0.001$. (C) Number of progenies with indicated mutations in F1 (Black dot), F2 (Blue square), and F3 (Red triangle) generations. The data shown are means±S.E.M. Asterisks indicate significant differences with One-way ANOVA test: ***$P<0.001$. (D) Representative immunofluorescence images of BmTSSKL in WT and *BmTSSKL* mutant virgin testes. TSSKL protein is red fluorescent, nucleus is blue fluorescence (Hoechst). Scale bar, 500 μm. (E-H) Response indices of control and *BmTSSKL* mutant. The data shown are means±S.E.M. (n=30/group). n. s. indicates no significance by a two-tailed *t*-test.

emerged male moths. The results showed that the sperm nuclei were assembled regularly at the head of the eupyrene sperm bundles in the WT, whereas in *BmTSSKL* mutants the eupyrene sperm bundles nuclei were abnormal positioning and exhibited abnormally squeezed eupyrene sperm bundles (Fig 3D). The apyrene sperm of WT had no polarity, and micronuclei were distributed in the middle region of bundles. In contrast, the apyrene sperm bundles in *BmTSSKL* mutants had defects in sperm nucleus shape and localization (Fig 3E). Thus, we deduced that a defect in *BmTSSKL* could influence both apyrene and eupyrene sperm morphogenesis.

### *BmTSSKL* mutation leads to mitochondrial derivative deficiency and activation of autophagy in dimorphic sperms

To explore further how *BmTSSKL* deficiency hinders sperm morphogenesis and sperm migration, we used transmission electron microscopy (TEM) to examine ultrastructural changes (mitochondria and microtubules) of apyrene and eupyrene sperm in the testes. We observed that the transverse sections of eupyrene sperm in WT showed that the elliptical mitochondrial derivatives were approximately equal in size and regularly arranged by side below the centrosome. Conversely, the mutant's mitochondria were not closely juxtaposed, and the sperm cell membrane ruptured, resulting in the extracellular dissociation of microtubule structure centrosome (Fig 4A). Importantly, we detected the elliptical mitochondrial derivatives of *BmTSSKL* mutants were markedly atrophied and disappeared, and surrounded by membrane-limited autophagosome vesicles (Fig 4A, *BmTSSKL-1* and *BmTSSKL-2*). The degraded vesicular material and mitochondrial remnants around disintegrated sperm structures were engulfed by autophagosomes (Fig 4A, *BmTSSKL-3*). In contrast, no autophagosomes were observed in eupyrene sperm WT testes. Moreover, we found the transverse sections of apyrene sperm showed that a pair of mitochondria of apyrene sperm present "V" shaped in WT, whereas, there were one, three or four mitochondrial derivatives in *BmTSSKL* mutant (Fig 4B). Collectively, TEM data indicated that mutation of *BmTSSKL* induced the production of defective mitochondrial derivatives and the activation of autophagy in dimorphic sperms.

To gain insights into changes in global gene expression associated with *BmTSSKL* deficiency, we performed RNA-Seq on sperm bundles isolated from testes of WT and *BmTSSKL* mutant at 2 days old virgin male adults. In a comparison between sperm in testes of *BmTSSKL* and WT, we identified 224 significantly upregulated and 59 downregulated genes (S2 Table). Remarkably, Kyoto Encyclopedia of Genes and Genomes (KEGG) analysis showed that the differentially-expressed genes (DEGs) were mainly in amino acid metabolism associated with synthesis and release of ATP (Fig 4C). We further verified the expression changes of genes involved in mitochondrial metabolism by qRT-PCR. The mRNA levels of *BmDlp1*, *BmFis1*, *BmMfpp1*, *BmGdap1*, *BmE3*, *BmMch*, *BmMib1*, and *BmSlp2* were significantly different in *BmTSSKL* mutants than those of WT (Fig 4D). The deletion of *BmTSSKL* gene led to the disturbance of sperm mitochondria fusion and fission, that is, the synthesis and release of ATP. This further affects genes related to sperm morphogenesis and motility, leading to male sterility.

### The functional conservation of *TSSKL* is widespread in Lepidotera

In order to further verify the evolutionary conservation of TSSKL in Lepidoptera, the *B. mori TSSKL* gene was used as a query to identify the putative coding sequence of *PxTSSKL* from the *Plutella xylostella* genome (S5 Fig). As expected, we observed that *PxTSSKL* is expressed at a higher level in the testis than other tissues of adult moths, and the *PxTSSKL*

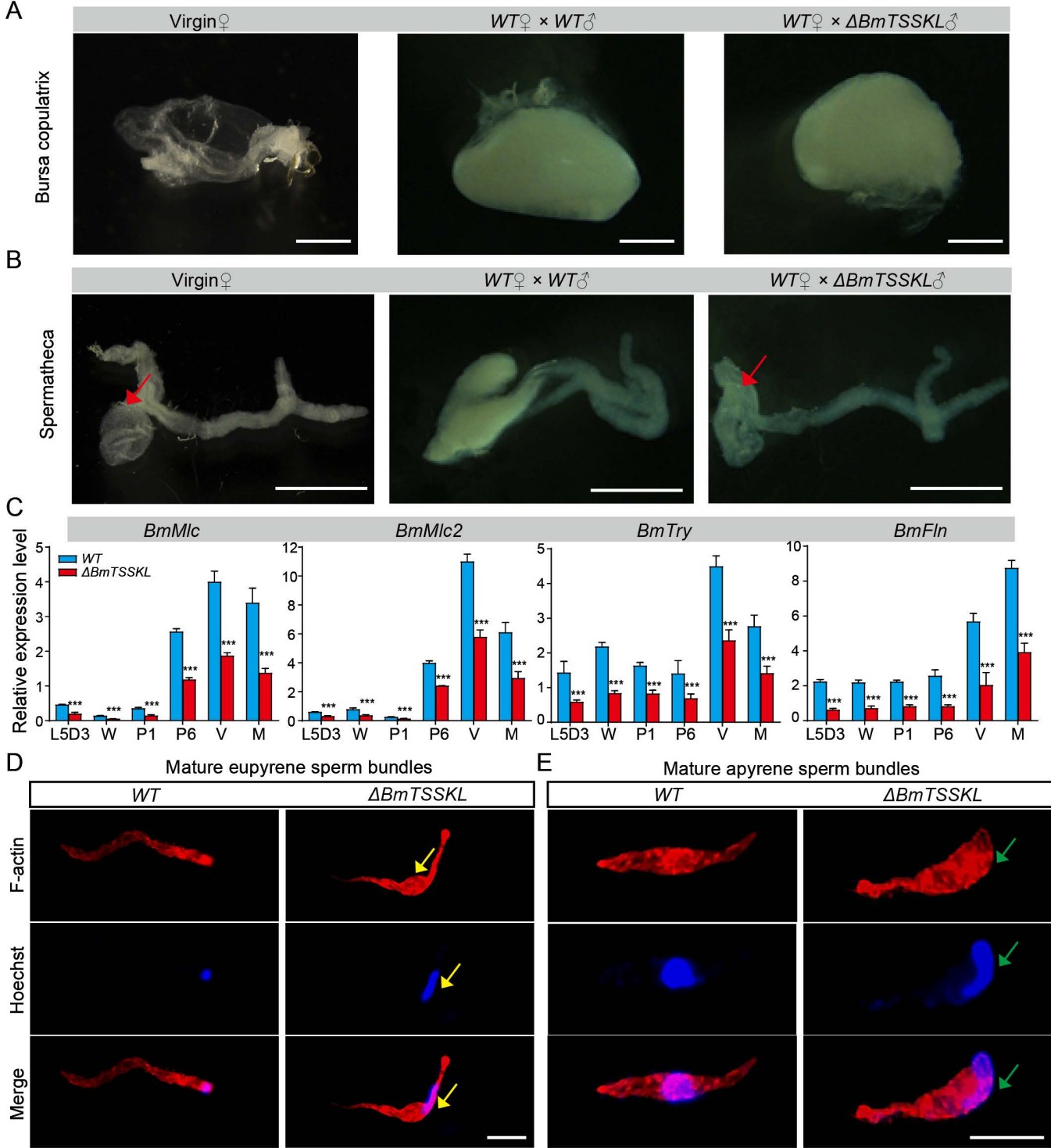

**Fig 3. *BmTSSKL* mutation affect sperm morphogenesis and sperm migration.** (A-B) Representative images of bursa copulatrix (A) and spermatheca (B) from three groups: virgin females, WT females mated with WT males, and WT females mated with *BmTSSKL* mutant males. Scale bar, 1 mm. (C) The mRNA expression of genes associated with sperm motility. The mRNA expression level was normalized to *B. mori ribosomal protein 49* (*Bmrp49*), an internal reference. Three independent replicates were performed. The data shown are means ± S.E.M. Asterisks indicate significant differences with a two-tailed *t*-test, ***$P<0.001$. (D-E) Representative immunofluorescence images of eupyrene sperm bundles (D) and apyrene sperm bundles (E) released from the testes of WT and *BmTSSKL* mutant males on the second day after eclosion. Red, F-actin. Blue, Hoechst. Scale bar, 100 μm.

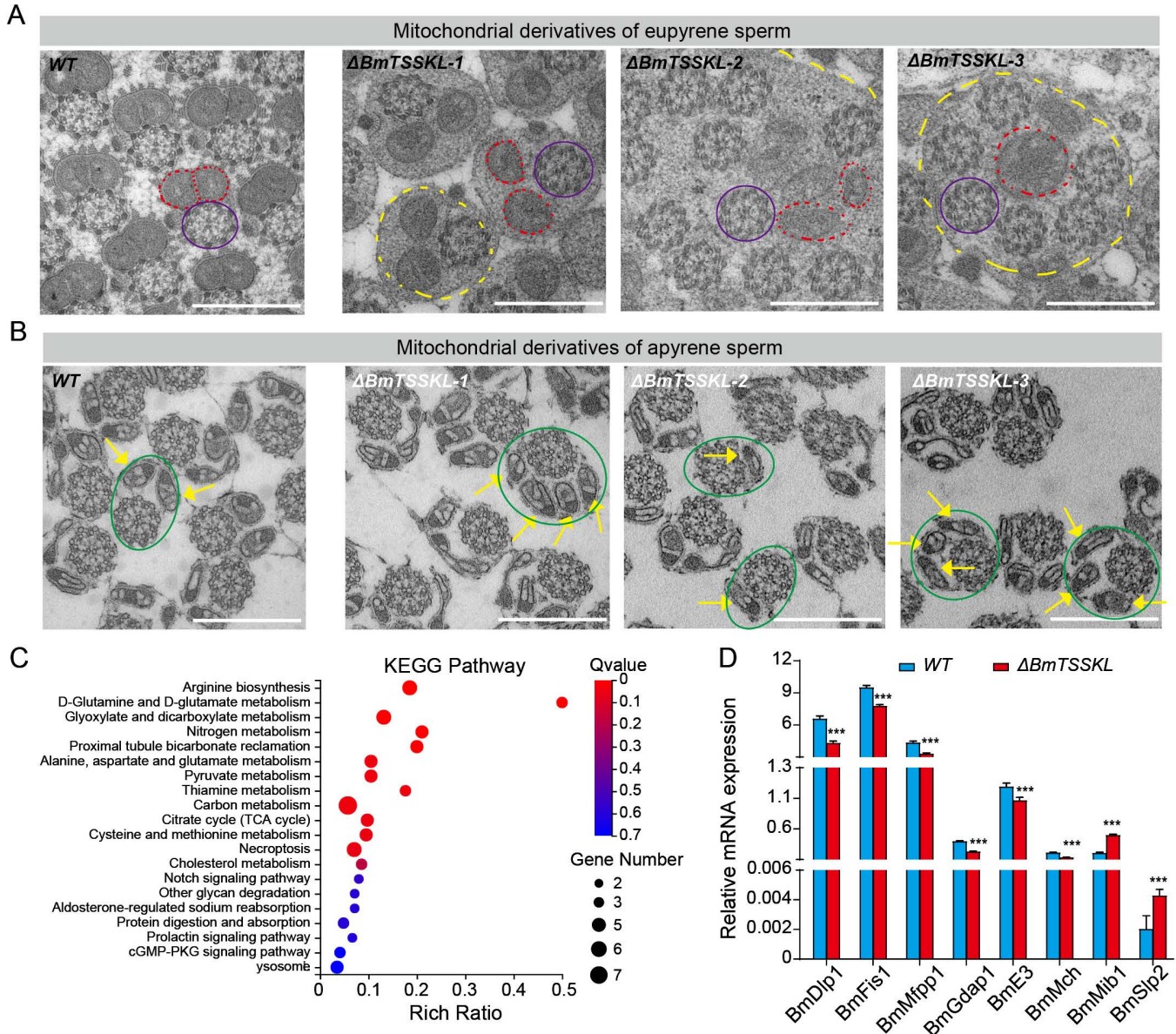

**Fig 4. CRISPR/Cas9-mediated deletion of *BmTSSKL* impaired mitochondrial derivatives and activated autophagy in sperm bundles.** (A) Representative transverse section images of eupyrene sperm bundles. Red dashed lines: mitochondrial derivatives. Purple ellipse lines: microtubule structure centrosome. Yellow dashed lines: autophagosome vesicles. Scale bar, 500 μm. (B) Representative transverse section images of apyrene sperm bundles. Green ellipse lines: microtubule structure and mitochondrial derivatives. Yellow arrows indicate individual mitochondrial derivatives of apyrene sperm bundles. Scale bar, 500 μm. (C) The top 20 enriched KEGG pathways of DEGs with $P < 0.05$ through RNA-Seq analysis in testes of WT and *BmTSSKL* mutant virgin adults. (D) Validation of RNA-Seq revealed gene expression changes in the mitochondria metabolic pathways by qRT-PCR. The mRNA expression level was normalized to *B. mori ribosomal protein 49* (*Bmrp49*), an internal reference. Three independent replicates were performed. The data shown are means ± S.E.M. Asterisks indicate significant differences with a two-tailed *t*-test, ***$P < 0.001$.

expression levels are higher in the testes of the 3-day-old pupal and virgin adult moths compared with other life stages (Fig 5A and 5B). Subsequently, we generated *PxTSSKL* mutants through CRISPR/Cas9 system in *P. xylostella* (Fig 5C and 5D). We found that *PxTSSKL* mutations don't affect the mating ability and mating behavior of females and males (S6 Fig). There was no significant difference in egg number among different cross combinations (Fig 5E and 5F).

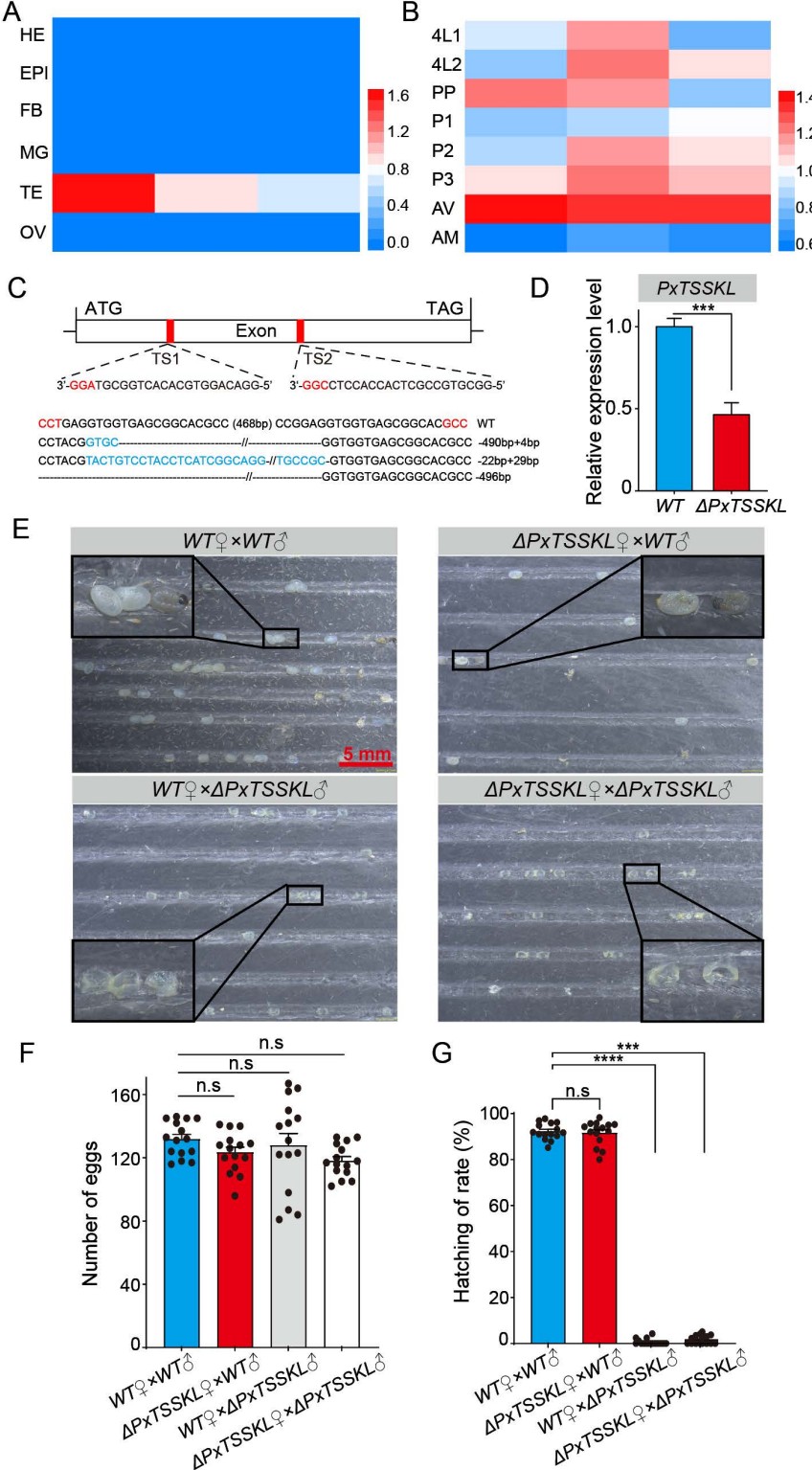

**Fig 5. *PxTSSKL* deficiency results male sterility in *P. xylostella*.** (A-B) Spatiotemporal expression pattern of *PxTSSKL*. HE, head; EPI, epidermis; FB, fat body; MG, midgut; TE, testis; OV, ovary. 4L1, the first day of the fourth instar larvae; PP, prepupa; P, pupae; AV, virgin adult; AM, mated adult. The mRNA expression level was normalized to *P. xylostella actin*, an internal reference. Three independent replicates were performed. (C) Schematic

of *PxTSSKL* gene structure and sgRNA target sites. The sequence of the region between sgRNA target sites in the *PxTSSKL* gene for WT and four randomly selected *PxTSSKL* mutant G0 individuals. Dashed lines represent the deleted bases and the blue characters represent the inserted bases. The net change in length is given to the right of each sequence (-, deletion; +, insertion). (D) Relative mRNA expression of *TSSKL* in WT and *PxTSSKL* mutant virgin testes. The mRNA expression level was normalized to *P. xylostella actin*, an internal reference. Three independent replicates were performed. (E) Representative photographs of eggs produced by different crosses. Eggs were photographed 2 days after spawning, 1 days before hatching. Developing eggs are dark and undeveloped embryos are shrunken. (F) The number of eggs laid for indicated crosses. (G) The hatching rate for indicated crosses. The data shown are means ± S.E.M. Asterisks indicate significant differences with One-way ANOVA test: ***$P < 0.001$, ****$P < 0.0001$, n. s, $P > 0.05$. Each group of (F) and (G) had at least five pairs of moths, and the experiments were performed three times (n > 15).

Nevertheless, we also obtained a mutant phenotype consistent with the results in silkworms where males are sterile and females are fertile (Fig 5E and 5G), suggesting that the *PxTSSKL* mutations cause male sterility in *P. xylostella*. Similarly, the deletion of *PxTSSKL* gene function resulted in abnormal sperm development in *P. xylostella*. Compared to the WT, the nucleus of eupyrene sperm could not be arranged in a needle pattern and became loose in mutant individuals compared to the WT (Fig 6A). The nucleus of apyrene sperm was excreted in WT individuals when it matured, while in *PxTSSKL* mutants, a large number of the nuclei were retained in apyrene sperm (Fig 6B). Meanwhile, we also observed bursa copulatrix was full in WT females mated with WT or *PxTSSKL* males (Fig 6C). Whereas, the spermathecae of WT females mated with *PxTSSKL* mutant males were empty (Fig 6D). Taken together, the resulted phenotypes in *PxTSSKL* mutants support the role of *TSSKL* in controlling sperm morphogenesis and motility in additional lepidopteran species.

## Discussion

Sperm morphogenesis represents a fundamental biological process with profound implications for reproductive success in sexually reproducing species [35]. Although TSSKs have been well-characterized as crucial regulators of sperm morphogenesis in model organisms [36,37], their functional significance in lepidopteran sperm dimorphism remains largely unexplored [38]. Here, we present the first systematic investigation of *TSSKL* function in Lepidoptera through comparative analysis of CRISPR/Cas9-generated mutants in both the model species *B. mori* and the agricultural pest *P. xylostella*. Our study reveals three key findings. First, *TSSKL* knockout induces complete male sterility while preserving normal mating behavior. Second, high-resolution ultrastructural analysis uncovers distinct mitochondrial defects in both sperm morphotypes: (i) eupyrene sperm display disrupted centrosomal mitochondrial pairing and exhibit autophagosome vesicles accumulation near mitochondria (Fig 7A and 7C), while (ii) apyrene sperm show aberrant mitochondrial derivative numbers (ranging from 1 to 4 instead of the wild-type 2) (Fig 7B and 7D). Third, these phenotypes are evolutionarily conserved between distantly related lepidopteran species. These results establish TSSKL as a key regulator of mitochondrial organization during lepidopteran sperm morphogenesis, providing new mechanistic insights into the maintenance of sperm dimorphism. Our findings bridge an important knowledge gap in comparative reproductive biology while identifying TSSKL-mediated mitochondrial dynamics as a potential target for species-specific fertility control.

Mitochondria, as highly dynamic organelles, maintain cellular homeostasis through precisely regulated fusion-fission cycles that govern their structural plasticity and functional specialization [39,40]. In spermatocytes, these processes are particularly crucial, as mitochondrial dysfunction is a well-established cause of male infertility across species, typically manifesting as spermatogenic arrest and defective sperm formation [41,42]. Our investigation of *TSSKL*-deficient lepidopterans revealed that *TSSKL* may play a role in maintaining mitochondrial-sperm relationships through two main mechanisms. First, *TSSKL* plays a central role in maintaining mitochondrial architecture through its dual regulation of organelle dynamics. Specifically, it coordinates mitochondrial fusion by modulating *Mch* and *Slp2* activity [43,44], while simultaneously controlling fission processes through *Dlp1*, *Fis1*, *Mfpp1*, and *Gdap1* [45–47]. Additionally, *TSSKL* fine-tunes mitochondrial quality control mechanisms by regulating the ubiquitination pathways mediated by *E3* and *Mib1* [48]. These regulatory functions exhibit distinct phenotypic consequences in the two sperm morphotypes: in apyrene sperm, *TSSKL* deficiency leads to structural disintegration characterized by disorganized mitochondrial derivatives and excessive autophagic activity (Fig 4A and 4B);

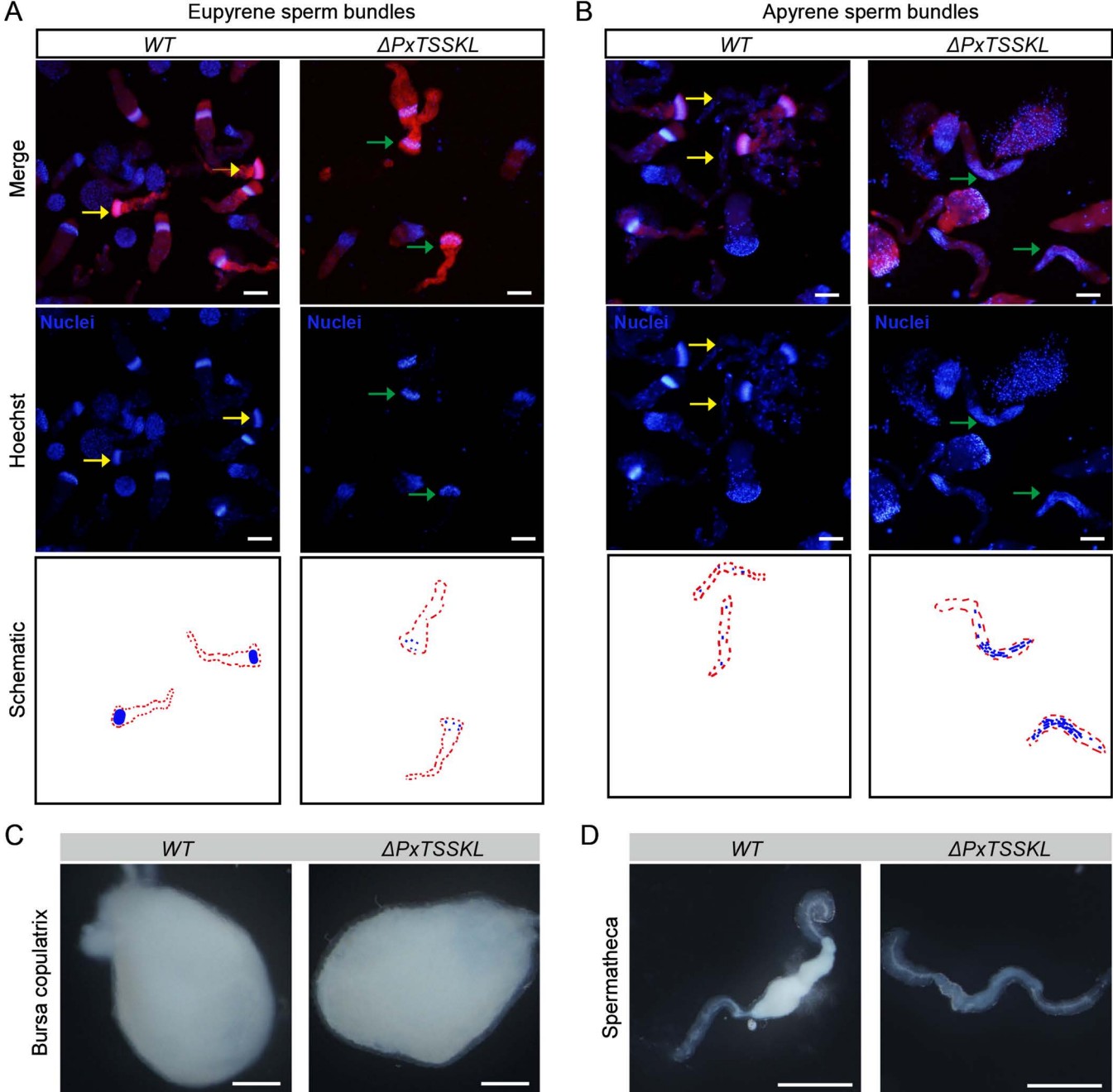

**Fig 6. Sperm morphology of virgin male testes.** (A) Representative immunofluorescence images of eupyrene sperm bundles. Yellow and green arrows indicate eupyrene sperm bundles. Red, F-actin. Blue, Hoechst. Scale bar, 50 μm. (B) Representative immunofluorescence images of apyrene sperm bundles. Yellow and green arrows indicate apyrene sperm bundles. Scale bar, 50 μm. Phalloidin was used to stain actin, and Hoechst was used to stain the nucleus. (C) Representative images of bursa copulatrix of WT females mated with WT or *PxTSSKL* mutant males. Scale bar, 200 μm. (D) Representative images of spermatheca of WT females mated with WT or *PxTSSKL* mutant males. Scale bar, 200 μm.

in contrast, eupyrene sperm primarily suffer from metabolic dysfunction, manifested as impaired oxidative phosphorylation resulting from the downregulation of *BmDlp1* and *BmFis1*. Second, *TSSKL* critically regulates mitochondrial bioenergetics by coordinating ATP synthesis pathways. Its knockout triggers a bioenergetic crisis through: i) impaired oxidative phosphorylation via *BmE3*/*BmMch* [43,48] dysfunction causing ATP depletion, and ii) cristae disorganization through *BmSlp2* deficiency [44]. These defects disrupt nuclear compaction and flagellar motility. Meanwhile, reduced *BmFln* and *BmMlc* expression impairs mitochondrial anchoring and flagellar function [31–34,49,50]. These data suggest that *BmTSSKL*-mediated sperm motility genes regulate the sperm to migrate from the bursa copulatrix to the spermathecae.

In summary, we observed mitochondrial abnormalities likely disrupt the precise ATP production required for sperm motility in *TSSKL* mutants, creating a cascade effect: mitochondrial dysfunction→compromised energy metabolism→defective sperm motility→complete male sterility (Fig 7). These findings highlight TSSKL's pivotal role in sperm

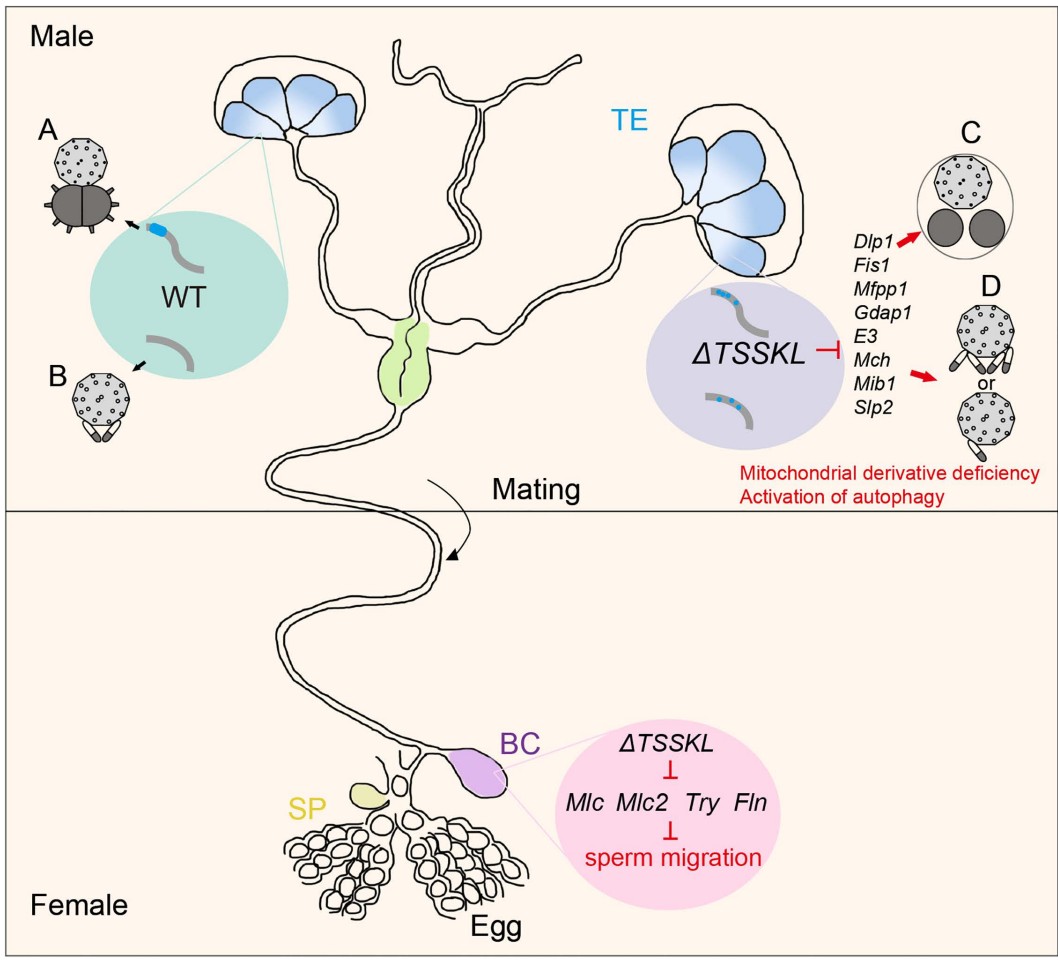

**Fig 7. Model illustrating *TSSKL* mediated regulation of male fertility in lepidopteran insects.** Sperm morphogenesis is normal in the reproductive system of WT (shown in green box). Eupyrene sperm bundles and apyrene sperm bundles in WT testes contain normal mitochondrial derivatives (A and B). In *TSSKL* mutant males, most of the eupyrene sperm bundles and apyrene sperm bundles had defects in sperm nucleus shape (purple box). Eupyrene sperm bundles and apyrene sperm bundles in *TSSKL* mutant testes contain abnormal mitochondrial derivatives (C and D). Compared with the WT moths, the autophagosome vesicles were formed close to the mitochondria as well as the structure mitochondrial derivatives of tended to be disintegrated (C). In the of females mated with *TSSKL* mutant males, almost no sperm is transferred from bursa copulatrix to the spermatheca for fertilization. Abbreviations: TE, testis; BC, bursa copulatrix; SP, spermatheca.

morphogenesis and a promising target for species-specific fertility control in lepidopteran pests. Nevertheless, the definitive roles of TSSKL, as a serine/threonine-protein kinase, and the definitive mechanistic interpretations remain open questions that deserve in-depth investigation.

## Methods

### Insect strains and rearing

Silkworm, *Nistari*, a multivoltine and non-diapausing strain, was used for all experiments. Larvae were reared on fresh mulberry leaves under standard conditions [51,52]. A laboratory strain of diamondback moth, *P. xylostella*, was obtained from Min-Sheng You (Fujian Agriculture and Forestry University, Fuzhou, China). Larvae were reared in an incubator at 26±1°C, 60±5% relative humidity, a photoperiod of 16:8 h (light: dark) and fed with an artificial diet as described previously [53]. Adult moths were fed with a 10% honey solution.

### RNA isolation, cDNA synthesis and qPCR analysis

Total RNAs of testis and other tissues were isolated from several tissues using TRIzol reagent (Invitrogen, USA). 1 µg of total RNA was used with the First Strand cDNA Synthesis Kit for complementary DNA (cDNA) synthesis (Promega, USA). Quantitative real-time PCR (qRT-PCR) analyses were performed using a SYBR Green Realtime PCR Master Mix (TOYOBO, Japan). The PCR conditions were as follows: initial incubation at 95°C for 5 min, 35 cycles at 95°C for 15 s, and 60°C for 1 min. The *B. mori* gene encoding ribosomal protein 49 (*Bmrp49*) and *P. xylostella Pxactin* were used as internal control. A relative quantitative method ($^{\triangle\triangle}$Ct) was used to evaluate quantitative variation. The gene-specific primers used for qRT-PCR are listed in S3 Table.

### Plasmid construction and germline transformation of silkworm

According to the construction principle of the CRISPR/Cas9 system, the open reading frames (ORFs) of *BmTSSKL* were screened to identify sgRNA target sites using the GGN19GG rule [54–56]. Two 23-base-pair (bp) sgRNAs that target sites in the exon of *BmTSSKL* were designed. The activator was the plasmid *pBac[IE1-GFP-Nos-Cas9]* (*Nos-Cas9*) with the enhanced green fluorescent protein (GFP) marker, which results in Cas9 expression that is driven by the *Nos* promoter [57]; The effector plasmid was *pBac[IE1-DsRed2-U6-sgRNAs]* (*U6-sgRNAs*) with the discosoma red fluorescent protein (dsRed) fluorescent marker, which results in expression of the sgRNAs under control of the silkworm *U6* promoter. Both GFP and dsRed fluorescence markers were controlled by the *IE1* promoter. The primers used for plasmid construction are listed in S3 Table.

To generate the *TSSKL* mutant, we carried out silkworm germline transformation by microinjecting *Nos-Cas9* and *U6-sgRNA* respectively into the pre-blastoderm *Nistari* embryos along with a mixture of *piggyBac* helper vectors. Then, embryos (generation 0: G0) were incubated in a humidified chamber at 25°C until hatching. The resulting larvae were reared to adulthood and backcrossed with WT moths to produce progeny. Through fluorescence microscopy screening, we selected G1 individuals displaying single fluorescent markers (GFP or DsRed) and intercrossed them to obtain G2 progeny. This crossing strategy yielded four distinct germlines: (1) dual fluorescent mutants (GFP/DsRed) expressing both Cas9 and sgRNA, (2) Nos-Cas9 line (GFP/+), (3) U6-sgRNA (DsRed/+), and (4) non-transgenic controls (fluorescence-negative) (S2 Fig). Meanwhile. the double-positive individuals (GFP/DsRed) were identified as putative mutants and selected for subsequent genotypic validation.

Inverse PCR was used to determine the genomic insertion sites of the transgene as described previously. In brief, Genomic DNA was extracted from G1 transgenic moths using the CTAB method. Subsequently, 5 µg of DNA was digested with DpnII, purified and self-circularized with T4 DNA ligase. Flanking genomic regions were amplified by nested PCR using outward-facing primers specific to the 5' transposon terminal repeat (S3 Table). The resulting PCR products were sequenced to identify the insertion sites.

## Knockout of *TSSKL* in *P. xylostella*

Two 23-bp sgRNA target sites were selected in exon 1 of *PxTSSKL* following GGN19GG principle. The sgRNAs were synthesized with a MegaScript T7 Kit (Thermo Fisher Scientific, USA). The primers used for plasmid construction are listed in S3 Table. A mixture of sgRNA (300ng/μL) and Cas9 proteins (300ng/μL) in 10 μL was injected into embryos within 2 hours using microinjectors. Subsequentlye, eggs were incubated in at $25 \pm 1°C$ and $65 \pm 5\%$ relative humidity until hatching, Individuals with *PxTSSKL* deletion were used in subsequent experiments.

## Mutagenesis analysis

Genomic DNA of individuals was extracted to identify deleted regions. First instar larvae were incubated in DNA extraction buffer (1:1:2:2.5 ratio of 10% SDS to 5 M NaCl to 100 mM EDTA to 500 mM Tris-HCl, pH = 8) with proteinase K (Thermo Fisher Scientific, USA), then purified with a standard phenol: chloroform extraction and isopropanol precipitation, followed by RNaseA (Takara, China) treatment. The genomic PCR conditions were as follows: 94°C for 2 min, 35 cycles of 94°C for 15 s, 55°C for 30 s, and 72°C for 1 min, followed by a final extension period at 72°C for 10 min. The PCR products were sub-cloned into pJET-1.2 vectors, and vectors were sequenced. The primers used for sequencing are listed in S3 Table.

## Mating ability and mutation transmission assay of silkworm

Silkworm female attraction and male competitiveness assay were performed in a box ($30 \times 18 \times 4.5 \, cm^3$). Newly emerged control and *BmTSSKL* mutant female moths were placed on either side of box, and WT male moth was placed in the center. Males reaching one side and mating with females were considered responsive, and the number of responding females was recorded. The response index was calculated as the number of reaction moths divided by the total number of test moths multiplied by 100. Using the same method of female attraction, statistical analysis of male competitiveness was carried out. The cross between *Nos-Cas9* line and *U6-sgRNA* line produced four lines, including *BmTSSKL* mutant line with double-fluorescent, *Nos-Cas9* line with green fluorescence, the *U6-sgRNA* line with red fluorescence, and a nonmutant line without fluorescence. Random hybridization of these four lines produced only these four types of individuals.

## Reproductive capacity assay of *P. xylostella*

Wild-type virgin females were allowed to mate with wild-type and *PxTSSKL* mutant males into a mesh cage ($25 \times 25 \times 25 \, cm^3$). Similarly, wild-type virgin males were allowed to mate with wild-type and *PxTSSKL* mutant females. All females were allowed to lay eggs for 24 h. The number of eggs laid by different groups of females and the hatching rate were analyzed. Each pair of mating males and females was placed separately in a clear tube to observe mating behavior and calculate offspring hatching rate. Each group had five pairs of moths, and the experiments were performed three times.

## Immunofluorescence analysis

Testes from individuals at the virgin adult stage were dissected and fixed in 4% paraformaldehyde overnight. They were dehydrated with different concentrations of ethanol and then embedded in paraffin. Samples were cut into ultrathin sections (4 μm), and dewaxed with dewaxing agent and ethanol. Subsequently, samples were immersed in a citric acid buffer for antigen repair. The rabbit anti-TSSKL primary antibody was used for TSSKL detection (1:500 dilution). Horseradish peroxidase-conjugated anti-rabbit IgG (1:1000 dilution) was used as a secondary antibody. The cell nucleus was re-stained with DAPI staining solution, and the film was sealed with anti-fluorescence attenuator. Image scanning through the scanner, and then observed by SlideViewer software.

PLOS Genetics

## Fluorescent staining

Testes from virgin male adults were dissected on an anatomical plate and placed in phosphate buffered saline (PBS). Each testis was torn in a dye fixing solution and fixed for 10 min, then washed three times with PBS, 5 min/ time. Phalloidin was used to stain actin in a 30 min incubation followed by three washes with PBS for 5 min each. Subsequently, nuclei were stained with Hoechst dye for 15 min, followed by washing 3 times with PBS for 5 min each, followed by transfer of the tissue in a droplet to a slide and covering with a cover glass. Images were captured using Olympus BX51 fluorescence microscope (Olympus, Japan).

## Transmission electron microscopy

Testes of virgin male adults were dissected and fixed with 2.5% glutaraldehyde in 0.1 M PBS overnight at 4°C. Samples were washed with PBS and fixed in the phosphate buffer with 1% osmium tetroxide for 2h. They were dehydrated with different concentrations of ethanol and then embedded in Embed812 resin. Samples were cut into ultrathin sections (60 nm), stained with 2% uranyl acetate (pH = 5) and then stained with 10 mM lead citrate (pH = 12). The stained sections were observed by Hitachi H-7650 transmission electron microscopy (Hitachi, Japan).

## RNA-seq analysis

Testes, ovary, and whole body of excluding the testis or ovary were dissected from 2 days old virgin adults. Then total RNA was extracted by using the RNAiso plus (Takara, China) following the manufacturer's protocol. Ten individuals were used for each sample, and three independent replicates were performed. Subsequently, total RNA was first enriched and then fragmented for cDNA synthesis and library construction. The DNBSEQ-T7 platform was used to sequence the library by BGI (https://www.genomics.cn/), and Fast QC was used to qualify the raw data, then the data were filtered by Trimmomatic. The filtered data were mapped and quantified to the reference *B. mori* database (https://www.ncbi.nlm.nih.gov/datasets/genome/GCF_030269925.1/). In addition, differentially expressed genes (DEGs) between WT and *BmTSSKL* mutant were normalized and analyzed by the DEGSeq R package (fold change >1 and FDR < 0.05 were used as a cutoff) [58]. Enrichment analyses of DEGs were conducted using the gene ontology (GO) analysis with all the genes which were expressed in our study as a background set [59]. All sequencing data have been deposited in GenBank under an accession code PRJNA1199782.

## Statistical analysis

All the samples have at least three independent replications in this study. Statistical analysis was performed using SPSS 23.0 software with an independent Student's *t*-test. The data were presented as means ± S.E.M, and statistically significant differences were represented by asterisks as *$P < 0.05$, **$P < 0.01$, ***$P < 0.001$.

## Supporting information

**S1 Fig. Genomic insertion loci of the *Nos-Cas9* (A) and *U6-sgRNAs* (B) transgenic lines.** The insertion sites are marked by vertical red arrows. The flanking genes are annotated with blue arrows, and the distances from the insertion sites to these genes are labeled in base pairs.
(TIF)

**S2 Fig. Mutation of *BmTSSKL* results in stable inheritance of male sterility.** Schematic of crosses done to demonstrate heritably of the male sterility phenotype in *B. mori*. The activator line (*Nos-Cas9*) is represented as a green moth, the effector line (*U6-sgRNAs*) is in red, the positive line is in half green and half red, and the negative line is in black. Four lines are produced by the hybridization of the *Nos-Cas9* line with the *U6-sgRNAs* line in *B. mori*.
(TIF)

**S3 Fig. Representative images showing morphological differences between WT and *BmTSSKL* mutants at key developmental stages: 3L1 (the first day of the third instar larvae), pupae, adult females and males.** Scale bars: 1 cm.
(TIF)

**S4 Fig. The representative sequencing chromatogram run from injected males and females.** Various deletions or insertions mutations were detected in injected males and females. Dashed lines represent the deleted bases, the green lowercase letters represent the inserted bases, and PAM are highlighted in red.
(TIF)

**S5 Fig. Protein sequence alignment of BmTSSKL from *B. mori* and PxTSSK from *P. xylostella*.**
(TIF)

**S6 Fig. Loss of *PxTSSKL* does not alter adult mating behavior in *P. xylostella*.** WT males mated with WT and *PxTSSKL* mutant females, *PxTSSKL* mutant males mated with WT and *PxTSSKL* mutant females. Scale bar, 5 mm.
(TIF)

**S1 Table. Specific expression of genes in the testis.**
(XLSX)

**S2 Table. Differentially expressed genes in the testes of WT and *TSSKL* mutants.**
(XLSX)

**S3 Table. Primers used in PCR amplification and plasmid construction.**
(XLSX)

**S1 Data. Raw data for figures.**
(XLSX)

## Author contributions

**Conceptualization:** Yongping Huang, Yaohui Wang.

**Data curation:** Xia Xu, Lu Zhu, Jine Chen, Xin Du, Linbao Zhu, Lijun Zhou.

**Formal analysis:** Xia Xu, Shaofang Yu, Lansa Qian, Yaohui Wang.

**Funding acquisition:** Xia Xu, Xingchuan Jiang, Yongcheng Dong, Yongqiang Wang, Yaohui Wang.

**Investigation:** Xia Xu, Lu Zhu, Xingchuan Jiang.

**Methodology:** Xia Xu, Lu Zhu, Lijun Zhou, Yongcheng Dong.

**Project administration:** Yongqiang Wang, Yongping Huang, Yaohui Wang.

**Resources:** Yongping Huang, Yaohui Wang.

**Software:** Xia Xu, Lansa Qian.

**Supervision:** Yongqiang Wang, Yongping Huang, Yaohui Wang.

**Validation:** Xia Xu, Lu Zhu, Xiaomiao Xu.

**Visualization:** Xia Xu, Yaohui Wang.

**Writing – original draft:** Xia Xu, Lu Zhu, Yaohui Wang.

**Writing – review & editing:** Yongqiang Wang, Yongping Huang, Yaohui Wang.

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
