## [Decision Letter · Decision Letter 0]

27 Jul 2025

PGENETICS-D-25-00718

TSSKL is essential for sperm mitochondrial morphogenesis and male fertility in moths

PLOS Genetics

Dear Dr. Wang,

Thank you very much for submitting your Research Article entitled, “TSSKL is essential for sperm mitochondrial morphogenesis and male fertility in moths” to PLOS Genetics. The manuscript was fully evaluated at the editorial level and by three independent peer reviewers. While the manuscript has merit and is of interest to PLOS Genetics, the reviewers raised some important concerns about the manuscript. In addition to addressing the point-by-point concerns of all reviewers, the following are critical changes/clarifications that must be made before the manuscript can be re-considered:

1.“Data not shown” statements are present throughout the text and this is not permitted. The authors either need to show the data supporting these conclusions or remove these claims entirely.

2. It is not entirely clear how the authors validated their mutants. They seem to be using a G0 CRISPR approach where the nos-cas9 x U63-gRNA-TSSKL progeny are used and called “mutants” but it does not appear that a single mutant KO line was derived for the experiments. The authors need to explicitly show genotyping information for the mutants and confirm that the female germline is also edited. Additionally, if G0s are being analyzed, the nos-cas9 would not drive KO anywhere but in the germ cells, so the authors cannot make conclusions about somatic roles of TSSKL. Several other related concerns were raised by the reviewers regarding the KOs. It is difficult to interpret the results without the nature of the mutations being made via CRISPR being defined and validated.

3. The number of eggs laid is not sufficient evidence to draw conclusions about female fertility in the mutants. The authors need to describe how they distinguish unhatched eggs from unfertilized eggs before inferring that the TSSKL KO impacts fertility.

4. Experimental replicates and n values are not included anywhere in the manuscript, and statistics are not sufficiently defined/explained.

5. The exact accession number / ID for the precise “BmTSSKL” in the study is not included anywhere.

6. There are general concerns about clarity and organization of the manuscript, including inconsistencies between the text and figures, and the Discussion section needs major revisions.

Should you decide to submit a revised version of the manuscript to PLOS Genetics for further consideration, please submit your revised manuscript within 60 days (before September 22, 2025). If you will need more time than this to complete your revisions, please reply to this message or contact the journal office at plosgenetics@plos.org. Please include the following items when submitting your revised manuscript:

We look forward to receiving your revised manuscript.

Kind regards,

Leah F. Rosin

Guest Editor

PLOS Genetics

Monica Colaiácovo

Section Editor

PLOS Genetics

Aimée Dudley

Editor-in-Chief

PLOS Genetics

Anne Goriely

Editor-in-Chief

PLOS Genetics

**Journal Requirements:**

At this stage, the following Authors/Authors require contributions: Xia Xu, Lu Zhu, Xiaomiao Xu, Jine Chen, Xin Du, Linbao Zhu, Shaofang Yu, Lansa Qian, Xingchuan Jiang, Lijun Zhou, Yongcheng Dong, Yongqiang Wang, Yongping Huang, and Yaohui Wang. Please ensure that the full contributions of each author are acknowledged in the "Add/Edit/Remove Authors" section of our submission form.

The list of CRediT author contributions may be found here: https://journals.plos.org/plosgenetics/s/authorship#loc-author-contributions

2) We noticed that you used the phrase 'data not shown' in the manuscript. We do not allow these references, as the PLOS data access policy requires that all data be either published with the manuscript or made available in a publicly accessible database. Please amend the supplementary material to include the referenced data or remove the references.

- ® on page: 20

- TM on page: 20.

Potential Copyright Issues:

i) Please confirm (a) that you are the photographer of 1A, 1G, 3A, and 3B, or (b) provide written permission from the photographer to publish the photo(s) under our CC BY 4.0 license.

ii) Figures 2E, and 7. Please confirm whether you drew the images / clip-art within the figure panels by hand. If you did not draw the images, please provide (a) a link to the source of the images or icons and their license / terms of use; or (b) written permission from the copyright holder to publish the images or icons under our CC BY 4.0 license. Alternatively, you may replace the images with open source alternatives. See these open source resources you may use to replace images / clip-art:

2) If any authors received a salary from any of your funders, please state which authors and which funders..

7)  Please ensure that the funders and grant numbers match between the Financial Disclosure field and the Funding Information tab in your submission form. Note that the funders must be provided in the same order in both places as well.  

**Reviewers' comments:**

Reviewer's Responses to Questions

**Comments to the Authors:**

Reviewer #1: In this study, Xu et al. utilized CRISPR/Cas9-mediated mutagenesis to characterize the functional role of the TSSKL gene in the lepidopteran insects. TSSKL knockout induced complete male sterility and severe morphological defects in both eupyrene and apyrene sperm. Comparative transcriptome and functional analyses linked these phenotypes to dysregulated energy metabolism pathways. Their results provided novel insights into insect reproductive biology.

The manuscript is concise, but sufficiently clear written. Experiments seem to be properly designed and precisely conducted. Results obtained are significant and well support conclusions made. I have not found any serious shortcomings, I suggest only some minor revisions.

Minor suggestions

Lines 164-166: BmTSSKL mutant males mated with WT or BmTSSKL mutant females produced eggs normally but did not hatch, whereas WT males mated with WT or BmTSSKL mutant females laid eggs and hatched normally (Fig 2A). This sentence seems to have a grammatical error.

Lines 212-214: we analyzed the mRNA expression of sperm motility genes at different developmental stages in WT and BmTSSKL mutant individuals was compared by using qRT-PCR. The meaning of this sentence seems unclear. Please check it.

For BmTSSKL, which number is the accession ID of this gene in GenBank. Please add the number in the manuscript.

In Supporting information, the file “S1 Table” was named as Table S2, “S2 Table” named as Table S3, and “S3 Table” named as Table S1. Please check them carefully.

Reviewer #2: In this manuscript, the authors elucidate the role of TSSKL in sperm mitochondrial morphogenesis in lepidopterans. Their findings demonstrate that TSSKL deficiency leads to severe morphological abnormalities in both eupyrene sperm and apyrene sperm, associated with disrupted mitochondrial dynamics and aberrant autophagy activation. Furthermore, RNA-seq analysis revealed impaired mitochondrial metabolism in the testes of TSSKL mutant males. Overall, this study provides novel mechanistic insights into insect reproductive genetics. However, several aspects of the manuscript require improvement, as outlined below:

1. The study reports consistent phenotypic effects in both Bombyx mori and Plutella xylostella. To strengthen the evolutionary context, please provide sequence alignment data.

2. For P. xylostella egg counting: Given the species' reproductive biology, please detail your quantification methodology and any statistical validation of sampling approaches.

3. Regarding mutagenesis strategies: Explain the rationale for using double mutants in B. mori versus single mutants in P. xylostella, including any technical constraints.

4. Expand the discussion on how TSSKL-mediated mitochondrial dysfunction specifically impacts both sperm types.

5. Discussion needs to be improved. Systematically integrate all examined genes into a proposed TSSKL regulatory network.

6. Figure 1B, In the selected tissues, why are OV, TE, EO and MO? Can the influence of other gonads be excluded?

7. Figure 2A and 5E, what is the explanation for the unhatched eggs in the mating between female mutants and wild-type individuals?

8. Figure 2C, I suggest asterisks be added to denote significance.

9. Figure 5G, the meaning of **** has not been explained.

Reviewer #3: This study by Xu et al., identified TSSKL as an essential regulator in sperm mitochondrial morphogenesis and male fertility of moths. Their genetic and molecular experiments are well designed, and the data are presented clearly and concisely. Altogether, this is a nice study that uses two species to demonstrate the conserved role of TSSKL in sperm functions. However, I have some comments and concerns that should be addressed.

1. Line 155: The authors state, “We confirmed that gene editing was successful by crossing Nos-Cas9 line with EGFP marker (Fig. 1G). However, it means that a Nos-Cas9/GFP line was used in combination with sgRNA/dsRed to induce TSSKL deletion or mutation. This strategy seems to serve as a method for generating or selecting TSSKL knockout/KO animals, rather than as a means of confirming successful gene editing.

2. Following Q1: by using this approach, would the authors acquire homozygote or heterozygote TSSKL mutants? Please provide some details to support ether way.

3. Line 168: Please include a description distinguishing undeveloped or unhatched eggs from those that developed/fertilized properly to better support the conclusion that infertility is caused by TSSKL knockout. The numbers of eggs won’t lead the conclusion of infertility caused by TSSKL KO.

4. Please provide numbers of biological replicates for all quantitative bar graph: e.g., Fig. 2B, 3C, Fig. 4. etc

5. Line 269: “…….were significantly difference in BmTSSKL mutant than those of WT” change to “……were significantly different in BmTSSKL mutants than those of WT.”

6. Please revise conclusions and discussion to precisely interpret your study:

1) Is Nos-CAS9 driven a Nanos/Nos promoter? If so, TSSKL deletion would be expected to occur specifically in germ cells, not in somatic cells. Thus, authors cannot draw such a conclusion that “TSSKL knockout induces complete male sterility while preserving normal somatic development and mating behavior, demonstrating its specific requirement in post-meiotic spermatogenesis” (Line 345), as the somatic effects were not directly assessed.

2) Were the germ cell progenitors affected by TSSKL KO? Without this information, the authors cannot conclude that TSSKL’s specific requirement in post-meiotic spermatogenesis.

3) The authors only presented the altered morphology of sperm but did not provide data about sperm counts in TSSKL KO. If only the functions (but not the development) of sperm were affected, the authors need to revise the conclusion that TSSKL regulates spermatogenesis.

4) While it is informative to get a global transcriptome changes resulting from TSSKL KO, it is important to recognize that TSSKL is a kinase, and its deficiency primarily affects protein modification or signaling pathways rather than directly regulating gene transcription. The observed alteration in gene transcription may therefore represent downstream or indirect effects of disrupted signaling networks or targets regulated by TSSKL. Indeed, several TSSK family members have been identified in mammals and are known to influence signal transduction of germ cells. The authors should consider these alternative mechanisms and revise their conclusion that 'Our studies reveal TSSKL’s central role in orchestrating mitochondrial dynamics, evidenced by its coordinated regulation of the core fusion/fission machinery.'"

5) The 2nd and 3rd paragraph of discussion section needs to be re-organized. The authors begin by discussing fusion and fission first, then shift to the topic to ATP synthesis and motor proteins, and then back to fusion and fission. This sequence lacks a clear logical flow and would benefit from a more cohesive structure.

**Have all data underlying the figures and results presented in the manuscript been provided?**

Reviewer #1: Yes

Reviewer #2: Yes

Reviewer #3: Yes

PLOS authors have the option to publish the peer review history of their article (what does this mean? ). If published, this will include your full peer review and any attached files.

**Do you want your identity to be public for this peer review?** For information about this choice, including consent withdrawal, please see our Privacy Policy .

Reviewer #1: No

Reviewer #2: **Yes: ** Muwang LI

Reviewer #3: No

**Figure resubmission:**
---

## [Editor Report · Decision Letter 1]

8 Sep 2025

PGENETICS-D-25-00718R1

TSSKL is essential for sperm mitochondrial morphogenesis and male fertility in moths

PLOS Genetics

Dear Dr. Wang,

Thank you for submitting your manuscript to PLOS Genetics. After careful consideration, we feel that it has merit but still does not fully meet PLOS Genetics's publication criteria as it currently stands. Therefore, we invite you to submit a revised version of the manuscript that addresses the points raised during the review process.

Please submit your revised manuscript within 30 days (10/13/25). If you will need more time than this to complete your revisions, please reply to this message or contact the journal office at plosgenetics@plos.org. Please include the following items when submitting your revised manuscript:

We look forward to receiving your revised manuscript.

Kind regards,

Leah F. Rosin, Ph.D.

Guest Editor

PLOS Genetics

Monica Colaiácovo

Section Editor

PLOS Genetics

Aimée Dudley

Editor-in-Chief

PLOS Genetics

Anne Goriely

Editor-in-Chief

PLOS Genetics

**Additional Editor Comments :**

While the manuscript is greatly improved and the authors have addressed many concerns raised by myself and the previous reviewers, I find that the rigor in this paper still falls short of a similar study previously published in PLOS genetics by this group. Some gaps in the manuscript that I believe still need to be addressed before publication are below.

1. The authors did not specify where and how many copies of the piggyBac cassettes are inserted, and how they are kept as stocks (single-insert or multicopy-insert). The best practice is to specify where the piggyBac cassette (e.g. with Splinkerette PCR) lands and keep the sgRNA and Cas9 stock as a single-copy stock. At the very least, the authors should explain the measures taken such that the transgenic individuals used for the subsequent sgRNA and Cas9 crosses do not have genes related to fertility-, sperm motility-, or mitochondria-related pathway already disrupted because of piggyBac random insertion.

2. For experimental hygiene, U6 and Cas9 transgenic line base fertility and responsive index will need to be compared to as control, instead of WT. A few experiments comparing the fertility and responsive index between WT and those lines will suffice to answer this question.

3. Line 153-154: Please elaborate how the dual fluorescent F2 and F3 are made to support “male sterility inheritance”. Maybe I misunderstood this claim or the schematic – but if F2 mutants are the result of G2 green and red (which is fertile x fertile), that is not male sterility inheritance.

4. Figure 3A: Please add a picture reference of virgin female’s bursa copulatrix and spermatheca.

5. Line 185-188: Please add references for these genes.

6. Line 189-191: The BmTSSKL mutant qPCR experiments show that the BmTSSKL gene KO affect the sperm-motility genes, not that these 4 genes affect migration. This claim may be more suitable to be extrapolated in the Discussion section.

7. Figure 5E: The authors need to define the measures taken to separate unmated eggs from mated eggs if Plutella cannot be hand-mated.

8. The authors show they can edit the female germline with no phenotype, so it puzzles me why they just didn’t recover a mutant from females to study. Please elaborate on this.

9. Line 237: typo “sperm sperm”.

**Journal Requirements:**

At this stage, the following Authors/Authors require contributions: Xia Xu, Lu Zhu, Xiaomiao Xu, Jine Chen, Xin Du, Linbao Zhu, Shaofang Yu, Lansa Qian, Xingchuan Jiang, Lijun Zhou, Yongcheng Dong, Yongqiang Wang, Yongping Huang, and Yaohui Wang. Please ensure that the full contributions of each author are acknowledged in the "Add/Edit/Remove Authors" section of our submission form.

The list of CRediT author contributions may be found here: https://journals.plos.org/plosgenetics/s/authorship#loc-author-contributions

2) We noticed that you used the phrase 'data not shown' in the manuscript. We do not allow these references, as the PLOS data access policy requires that all data be either published with the manuscript or made available in a publicly accessible database. Please amend the supplementary material to include the referenced data or remove the references.

3) We have noticed that you have uploaded Supporting Information files, but you have not included a complete list of legends. Please add a full list of legends for your Supporting Information file (Raw data for Figure.xlsx) after the references list.

**Figure resubmission:**
---

## [Editor Report · Decision Letter 2]

10 Oct 2025

Dear Dr Wang,

We are pleased to inform you that your manuscript entitled "TSSKL is essential for sperm mitochondrial morphogenesis and male fertility in moths" has been editorially accepted for publication in PLOS Genetics. Congratulations!

Yours sincerely,

Monica Colaiácovo

Section Editor

PLOS Genetics

Aimée Dudley

Editor-in-Chief

PLOS Genetics

Anne Goriely

Editor-in-Chief

PLOS Genetics

BlueSky: @plos.bsky.social

Comments from the reviewers (if applicable):

**Data Deposition**

http://datadryad.org/submit?journalID=pgenetics&manu=PGENETICS-D-25-00718R2

**Press Queries**

---

## [Editor Report · Acceptance letter]

PGENETICS-D-25-00718R2

TSSKL is essential for sperm mitochondrial morphogenesis and male fertility in moths

Dear Dr Wang,

We are pleased to inform you that your manuscript entitled "TSSKL is essential for sperm mitochondrial morphogenesis and male fertility in moths" has been formally accepted for publication in PLOS Genetics! Your manuscript is now with our production department and you will be notified of the publication date in due course.

With kind regards,

Lilla Horvath

PLOS Genetics

On behalf of:
